# Clinical Phenotypes of Parkinson’s Disease Associate with Distinct Gut Microbiota and Metabolome Enterotypes

**DOI:** 10.3390/biom11020144

**Published:** 2021-01-22

**Authors:** Sarah Vascellari, Marta Melis, Vanessa Palmas, Silvia Pisanu, Alessandra Serra, Daniela Perra, Maria Laura Santoru, Valentina Oppo, Roberto Cusano, Paolo Uva, Luigi Atzori, Micaela Morelli, Giovanni Cossu, Aldo Manzin

**Affiliations:** 1Department of Biomedical Sciences, University of Cagliari, 09042 Monserrato, Italy; vanessapalmas@hotmail.it (V.P.); s.pisanu6@gmail.com (S.P.); aleserra@unica.it (A.S.); daniperra@unica.it (D.P.); marialaurasantoru@gmail.com (M.L.S.); latzori@unica.it (L.A.); morelli@unica.it (M.M.); aldo.manzin@gmail.com (A.M.); 2Department of Neurology, Azienda Ospedaliero Universitaria, Policlinico ‘D. Casula’, 09042 Monserrato, Italy; mrtmelis@gmail.com; 3Department of Neurology Azienda Ospedaliera Brotzu, 09134 Cagliari, Italy; valentinaoppo@gmail.com; 4CRS4, Science and Technology Park Polaris, Piscina Manna, 09010 Pula, Italy; robycuso@crs4.it; 5IRCCS Giannina Gaslini Institute, 16147 Genova, Italy; paolouva@gaslini.org

**Keywords:** parkinson’s disease, clinical phenotype, gut microbiota, metabolome

## Abstract

Parkinson’s disease (PD) is a clinically heterogenic disorder characterized by distinct clinical entities. Most studies on motor deficits dichotomize PD into tremor dominant (TD) or non-tremor dominant (non-TD) with akinetic-rigid features (AR). Different pathophysiological mechanisms may affect the onset of motor manifestations. Recent studies have suggested that gut microbes may be involved in PD pathogenesis. The aim of this study was to investigate the gut microbiota and metabolome composition in PD patients in relation to TD and non-TD phenotypes. In order to address this issue, gut microbiota and the metabolome structure of PD patients were determined from faecal samples using 16S next generation sequencing and gas chromatography–mass spectrometry approaches. The results showed a reduction in the relative abundance of Lachnospiraceae, *Blautia*, *Coprococcus*, *Lachnospira*, and an increase in Enterobacteriaceae, *Escherichia* and *Serratia* linked to non-TD subtypes. Moreover, the levels of important molecules (i.e., nicotinic acid, cadaverine, glucuronic acid) were altered in relation to the severity of phenotype. We hypothesize that the microbiota/metabolome enterotypes associated to non-TD subtypes may favor the development of gut inflammatory environment and gastrointestinal dysfunctions and therefore a more severe α-synucleinopathy. This study adds important information to PD pathogenesis and emphasizes the potential pathophysiological link between gut microbiota/metabolites and PD motor subtypes.

## 1. Introduction

The pathological characteristics of PD consist of the inclusion of intra-neuronal α-synuclein, also called Lewy bodies, and dopaminergic neuron loss in the substantia nigra pars compacta [1]. The Lewy pathology and the loss of dopaminergic neurons are responsible for the distinctive clinical manifestations of the disease. PD is associated with a heterogeneity of clinical changes, suggesting the existence of different subtypes, characterized by both motor and non-motor dysfunctions [2,3]. Although it remains to be established whether the different subtypes represent distinct disorders or different disease stages [4], a current classification based on motor signs suggests two main subtypes of tremor dominant (TD) and non-tremor dominant (non-TD) [2,5]. Typically, in the TD subtype tremors are the main motor feature [6,7,8]. On the other hand, non-TD subtypes may display an akinetic-rigid syndrome characterized by rigidity, bradykinesia and akinesia, as well as an increased occurrence of postural instability and non-motor features (akinetic-rigid (AR) phenotype) [5,9,10]. As the disease progresses, motor symptoms worsen with the onset of further complications, particularly motor fluctuations and dyskinesia [11,12], usually associated with prolonged levodopa administration. Interestingly, one subgroup of patients are particularly prone to developing dyskinetic symptoms at a relatively early stage of the disease (D phenotype). Such phenotype is less likely to appear in TD patients [13]. It is important to notice that the onset of motor symptoms in PD is preceded by pre-motor manifestations, such as gastrointestinal (GI) dysfunctions, including constipation, prolonged intestinal transit time and pathophysiological changes in the intestinal barrier [14,15,16,17,18,19,20,21,22,23]. Previous studies reported that these changes can be associated with modification of gut microbiota composition and microbial metabolites [24]. Though it has been proposed that the different expressions of motor symptoms may reflect distinct pathophysiological mechanisms, emerging evidence from an animal model of PD suggests that the gut microbiota itself may promote neurodegenerative changes [25,26,27] and impair motor function through microglial activation [28].

Although the origin of Lewy pathology is still unclear, one of the most relevant theories, Braak’s hypothesis, has suggested that an injury of the enteric nervous system (ENS) triggered by gut bacteria may promote α-synuclein aggregation and its spreading, via the vagal nerve, from the ENS to the brain through the brainstem, midbrain, basal forebrain and the cortical areas [28,29,30]. On the other hand, different lines of evidence have suggested that α-synucleinopathy could originate in the olfactory bulb or in the brain, and then spread to the peripheral autonomic nervous system [29]. 

To date, several studies have investigated the gut microbial changes associated with PD patients [30,31,32,33,34,35,36,37,38]; however, several points of interest remain to be established. One of these concerns whether the dopaminergic drugs can affect gut bacteria or vice versa. Recent studies highlighted that gut microbes can affect the metabolism of L-DOPA, the most effective anti-Parkinson medication, potentially reducing its effectiveness and therefore the clinical manifestations of the disease [39,40,41]. Another point of interest that is currently quite conflicting, concerns the correlation between specific groups of bacteria with TD or non-TD phenotypes [36,42,43].

Based on the early development of the GI dysfunction and the evidence that the intestinal microbiota impacts brain functions with potential pathophysiological effects in motor impairment in PD, we investigated whether variations in the composition of gut microbiota/metabolome may be associated to different TD and non-TD motor subtypes of PD, in particular TD versus AR and D phenotypes.

## 2. Materials and Method

### 2.1. Patients and Clinical Assessment

All patients provided written informed consent after the study was approved by the local Institutional Ethics Committee (Prot.PG/2017/17817) of the Azienda Opedaliero Universitaria di Cagliari, Italy. Idiopathic PD patients (*n* = 56) diagnosed according to the UK Brain Bank criteria were recruited at the Neurology Department AO Brotzu (Cagliari, Italy). 

All PD were evaluated by the Movement Disorder Society-Unified Parkinson’s Disease Rating Scale (MDS-UPDRS https://www.movementdisorders.org/MDS/MDS-Rating-Scales/MDS-Unified-Parkinsons-Disease-Rating-Scale-MDS-UPDRS.htm) part III and IV and by the Non-Motor Symptom Scale (NMSS) and were classified into two main groups according to phenotype categories: tremor dominant (TD) (*n* = 19) and akinetic rigid (AR) (*n* = 23). We have also included another phenotypic group in the study called Diskinetic (D, *n* = 14). Patients displaying prominent dyskinesias were defined as “dyskinetic”; this definition was applied to those who had a score of at least 2 over 4 in the sequences 2, 4.3, 4.4 in the MDS-UPDRS part IV and/or onset of dyskinesias within 3 years from the start of therapy with levodopa. 

This criterion was selected because up to 50% of PD patients can develop dyskinesias and motor fluctuations after three/four years of levodopa treatment; this effect is the clinical reflection of a breakdown of neuronal homeostasis in the central nervous system [44]. Therefore, we decided that the onset of dyskinesias after less than three years of exposure to levodopa reflected a phenotype specifically prone to this type of disabling complication. All patients were enrolled on the basis of a well-balanced Mediterranean diet without nutritional variations, as defined in the Mediterranean diet questionnaire [45]. The exclusion criteria were: atypical parkinsonism; neurological or psychiatric illness; severe cognitive impairment; primary gastrointestinal disease; simultaneous presence of internal medicine; use of probiotics or antibiotics in the 3 months before enrolment. All patients included in the study were given levodopa-carbidopa (LD) for at least 3 years (short LD) or more (long-term LD), or alternatively LD-carbidopa intestinal gel (LCIG). Patients did not assume inhibitors of catechol-o-methyl-transferase.

### 2.2. Gut Microbiota Analysis

Gut microbiota composition in PD patients with different phenotypes (TD; RA; D) was investigated. DNA samples from frozen stools were extracted and analyzed by 16S rRNA sequencing, as previously described [38,46]. A total of 56 samples were sequenced using an Illumina MiSeq platform. Sequencing data were deposited in the European Nucleotide Archive (https://www.ebi.ac.uk/ena), under the study accession numbers PRJEB30401. 

### 2.3. Gut Metabolome Analysis

Gut metabolome analysis was carried out as previously described [38]. In brief, faecal metabolites were extracted from each sample in methanol solution. One microliter of derivatized sample was injected into a 7890A gas chromatograph coupled with a 5975C Network mass spectrometer (Agilent Technologies, Santa Clara, CA, USA) equipped with a fused silica capillary column for gas chromatography–mass spectrometry (GC-MS) analysis. The gas flow rate through the column was 1 mL/min. Relative concentrations of the discriminant metabolites were determined by the chromatogram area and then normalized.

### 2.4. Data and Statistical Analysis

The variations in the frequency of phenotype groups were determined using Pearson’s chi-squared test. 

Analysis of the data generated on the Miseq System platform was carried out as previously described [38] using the BaseSpace 16S Metagenomics App (Illumina). MicrobiomeAnalyst tool [47] was used to estimate alpha- and beta-diversity indices. The linear discriminant analysis effect size (LEfSe) (http://huttenhower.sph.harvard.edu/galaxy/) and the non-parametric factorial Kruskal–Wallis sum rank test, followed by the Benjamini and Hochberg false discovery rate (FDR) test for multiple comparisons was carried out to detect bacterial taxa that were statistically different among PD phenotypes groups. 

The general linear model (GLM) analysis and Bonferroni correction test for multiple comparisons were used on Statistical Package for the Social Sciences (SPSS) (version 25.0 for Windows) to evaluate the effect of potential confounders (sex, age, BMI, coffee consumption, smoking status and pharmacological treatment covariates) on gut microbiota composition related to PD phenotypes.

For the metabolome analysis the multivariate statistical analysis was carried out as previously described [38] using SIMCA-P software (version 14.0, Umetrics, Sweden). To evaluate the significant differences of metabolites among the phenotype groups a Mann–Whitney U test followed by the Holm–Bonferroni correction test were used.

## 3. Results

### 3.1. PD Patient Subtypes

The distribution of the different clinical phenotypes was determined in the PD patient cohort (Table 1). 

PD patients with the TD phenotype were 33.93% (*n* = 19), while 41.07% (*n* = 23) presented AR phenotype and 25.00% presented D phenotype (*n* = 14). No significant variations were observed in the distribution between TD vs. AR and D phenotypes groups (χ^2^ = 0.609, *p* = 0.434; χ^2^ = 1.066, *p* = 0.301), and between AR and D phenotypes (χ^2^ = 3.269; *p* = 0.070).

### 3.2. Demographic and Clinical Characteristics

The demographic and clinical characteristics of PD patients in regard to clinical phenotypes are shown in Table 2. 

The phenotype groups were comparable for most of the considered variables, such as age, gender, BMI, constipation, coffee consumption and smoking status. 

### 3.3. Changes in Bacterial Diversity Associate with Distinct PD Phenotypes

The variation of microbial communities within sample (alpha-diversity) and between samples (beta-diversity) was estimated in the PD phenotype groups. Marked differences in alpha-diversity were obtained in the comparison between the TD phenotype vs. AR and D phenotypes (Figure 1a–d). 

Significant differences were observed among Abundance-based Coverage Estimator ACE, Chao, Fisher, Shannon indexes (*p* values from ≤ 0.001 to ≥ 0.032); however, the Simpson index was not significantly different in comparison to the TD and D phenotypes, or the Shannon and Simpson indexes between TD and AR phenotypes (*p* values ≥ 0.05). The TD group showed the highest alpha-diversity, while on the contrary D and AR revealed the lowest alpha-diversity. No differences in alpha diversity analysis were detected in the comparison between AR and D phenotypes. Similarly, the inter-sample beta-diversity analysis showed the separation between the TD phenotype vs. AR and D phenotypes groups (*p* value = 0.003; *p* value = 0.001, respectively) (Figure 2a–b), while no significant differences were found between the AR and D phenotypes (*p* values > 0.05) (data not shown).

### 3.4. PD Phenotypes Differs in the Taxonomic Composition of Gut Microbiota 

The changes in the gut microbiota composition related to PD phenotypes are shown in Table 3 and Figure 3a–c. The main modifications included 16 taxa that were significantly different in comparison between the TD and AR phenotypes (Table 3, Figure 3a). 

The relative abundance of the Firmicutes phylum and several families such as Clostridiaceae, Gemellaceae and Lachnospiraceae was decreased in the AR phenotype. A reduction in the Brevibacteriaceae family within the Actinobacteria phylum was also observed. At a genus level, the most relevant reductions concerned *Brevibacterium* within the Brevibacteriaceae family, *Tindallia* within the Clostridiaceae family, *Gemella* within the Gemellaceae family and *Blautia*, *Coprococcus* and *Lachnospira* within the Lachnospiraceae family. The relative abundance of *Faecalibacterium* within the Ruminococcaceae family showed a decreased abundance in the same group as well. On the contrary, an increase in the Enterobacteriaceae family and *Escherichia* and *Serratia* genera and *Sedimentibacter* within the Peptostreptococcaceae family was observed.

Similar changes were documented in the comparison between D vs. TD phenotypes: 22 taxa were significantly modified (Table 3, Figure 3b). At the phylum level only Firmicutes decreased in the D phenotype. At the family level, the most relevant reductions concerned the Brevibacteriaceae, Clostridiaceae, Eubacteriaceae, Gemellaceae, Lachnospiraceae, Peptococcaceae and Ruminococcaceae families. A reduction in the abundance of several genera, such as *Brevibacterium*, *Tindallia, Acetobacterium*, *Gemella, Blautia*, *Coprococcus, Lachnospira, Faecalibacterium* and *Sedimentibacter* was also observed. On the contrary, the relative abundance of the Lactobacillaceae family and related genus *Lactobacillus*, and the Enterobacteriaceae family and *Escherichia* and *Serratia* genera increased in D phenotype.

The relative abundance of only two taxa, the Lactobacillaceae family and related genus, increased in the D phenotype compared to the AR phenotype (Table 3, Figure 3c).

Most differences in the gut microbiota composition among PD phenotypes groups were maintained at various taxonomic levels when the same data were corrected for several confounding factors, such as sex, age, BMI, coffee consumption, smoking status and pharmacological treatments by analysis of covariance (ANCOVA) (Table 3, * adjusted *p* values <0.05). The significant differences still associated with the AR phenotype, compared to the TD phenotype, consisted of a reduction in the Brevibacteriaceae and Lachnospiraceae families and several genera, such as *Brevibacterium*, *Blautia*, *Coprococcus* and *Lachnospira.* In addition, a significant increase in Enterobacteriaceae and related genus *Serratia,* and *Sedimentibacter* was also still observed. In the comparison between D vs. TD phenotypes, the significant differences concerned a reduction in several families, such as Brevibacteriaceae, Eubacteriaceae, Gemellaceae, Lachnospiraceae, and Peptococcaceae. The depletion of several genera such as *Brevibacterium*, *Acetobacterium*, *Gemella, Blautia*, *Coprococcus, Lachnospira,* and *Sedimentibacter* was still associated with the D phenotype, while increases were noted for *Escherichia* and *Serratia* genera. No significant differences in the microbiota composition were preserved when the D phenotype group was compared to the AR phenotypes ((−) *p* value > 0.05).

### 3.5. Distinct Metabolic Changes Associate with PD Phenotypes 

The multivariate statistical analysis (MVA) related to the gut metabolome composition among PD phenotypes groups is shown in Figure 4a–c.

The quality parameters of the orthogonal partial least-square discriminant analysis (OPLS-DA) model and the permutation test showed the statistical validity of the analysis between AR vs. TD phenotypes (R2Y, 0.590; Q2, −0.155; R2 intercept, 0.0, 0.662; Q2 intercept, 0.0, −0.137), D vs. TD phenotypes (R2Y, 0.757; Q2, 0.249; R2 intercept, 0.0, 0.718; Q2 intercept, 0.0, −0.133) and D vs. RA phenotypes (R2Y, 0.661; Q2, −0.210; R2 intercept, 0.0, 0.628; Q2 intercept, 0.0, 0.072). The results revealed distinct metabolic changes associated with the PD phenotypes (Figure 5a–c). 

An up-regulation of cadaverine and glucuronic acid was observed in the comparison between D phenotype vs. TD and AR phenotypes (FDR corrected *p* values < 0.05) (Figure 5a,c). On the contrary, a downregulation of nicotinic acid was observed in the same comparisons, D phenotype vs. TD and AR phenotypes (FDR corrected *p* values < 0.05) (Figure 5b).

## 4. Discussion

Our data highlight that the gut microbiota and metabolome composition differ in PD patients in relation to the clinical phenotypes. This work provides important information that better defines the potential pathological interaction between gut microbes–bacteria metabolites and motor PD subtypes. The results revealed that the overall gut microbiota structure showed a higher diversity and richness of bacterial species associated to the TD form, whereas a reduction in diversity and richness linked to non-TD phenotypes was also apparent. The beta diversity analysis showed that the composition of gut microbial communities was similar between PD patients with AR and D phenotypes; however, both non-TD phenotypes differed from the TD phenotype. While a reduction in the abundance of different taxa, such as Eubacteriaceae and related genus *Acetobacterium,* Gemellaceae and *Gemella* genus, and Peptococcaceae family, was exclusively related to the D form, the main relevant differences in the composition of gut microbiota were common in both the non-TD phenotypes. In particular, an increase in the Enterobacteriaceae family within the Proteobacteria phylum and related genus *Escherichia* and *Serratia* was linked to non-TD phenotypes. It has been suggested that enrichment of Enterobacteriaceae and key members plays an important role in PD dysbiosis [35,37]. Indeed, the gut microbiota dysbiosis is characterized by a shift in relative bacterial abundances with a prevalence of pathobionts belonging to the phylum Proteobacteria (i.e., Enterobacteriaceae), while beneficial symbionts belong mainly to the phylum Firmicutes and Bacteroides are less expressed. This persistent imbalance of the gut microbial community between harming and non-harming symbionts induces an immune reaction, which promotes an inflammation status that represents a favorable microenvironment for the growth of the same Enterobacteriaceae [48].

Interestingly, Enterobacteriaceae are among the most typically overgrown symbionts in many conditions implying inflammation, such as inflammatory bowel disease (Crohn’s disease and ulcerative colitis), obesity and colorectal cancer [48]. The overgrowth of Enterobacteriaceae in the gut has been associated with oxidative stress and alteration of barrier integrity due to continuous exposition to bacterial endotoxins [48]. Moreover, it is known that an inflammatory environment and an increase in gut permeability, triggered by bacterial endotoxins, can lead to an enhancement of α-synuclein expression and aggregation in the ENS [20,49,50]. In fact, it has been reported that an increase in α-synuclein aggregation may further activate microglia, which leads to additional α-synuclein propagation and progression of the disease [51]. Our findings are in agreement with a previous report [36] that showed an increase in Enterobacteriaceae in patients affected by postural instability disorder with gait difficulty (PIGD phenotype; a non-TD form related to the AR phenotype). In addition to the aforementioned enrichment of Enterobacteriaceae, our results strengthen the association between dysbiosis and non-TD phenotypes, extending the previous report with findings of a parallel reduction in several butyrate-producing bacteria within the Firmicutes phylum, as well as the Lachnospiraceae family and some genera in the same family, such as *Blautia*, *Coprococcus,* and *Lachnospira*. It is known that butyrate-producing bacteria are involved in promoting gastrointestinal integrity and motility [52] and in the modulation of intestinal inflammation by activation of the G protein-coupled receptor GPR109A [53]. Interestingly, it has been reported that subjects with PD display intestinal inflammation [21]. These considerations allow us to hypothesize that in PD patients with non-TD phenotypes, a co-reduction in butyrate-producing bacteria (i.e., Lachospiraceae members), known to be associated with PD [32,34,37,38], may impact gut homeostasis even more, thus, exacerbating the inflammation and GI dysfunctions. Based on this knowledge, the results from the present study suggest that a greater abundance of pro-inflammatory Enterobacteriaceae and a reduction in protective Lachospiraceae members in non-TD patients might be associated with gut inflammation and more severe α-synucleinopathy in the ENS. It has been proposed that non-TD phenotypes might be associated with a worse prognosis and faster progression of PD compared to TD patients and also with a more severe colonic α-synucleinpathology [22,36,54]. Even though more genetic and neuropathological studies are needed to establish the relationship of the gut microbiota changes with α-synuclein, leucine-rich repeat kinase 2, or glucocerebrosidase gene expression, as well as with other proteinopathies (i.e., amyloid-beta, tau and TAR DNA binding protein 43), our results suggest that gut dysbiosis may play a role in the different natural history and prognosis of disease related to TD and non-TD phenotypes. 

Although the analysis of faecal metabolites did not reveal significant differences in the total levels of butyrate between TD and non-TD phenotypes, the metabolome findings in our study are consistent with the pro-inflammatory microbiota profile associated with non-TD subtype. Specifically, depletion of nicotinic acid and a parallel increase in cadaverine and glucuronic acid is associated with non-TD (D phenotype) phenotypes, instead of the TD and AR phenotypes. It has been reported that nicotinic acid (vitamin B3) shares the same butyrate receptor in the gut [53], GPR109A, displaying anti-inflammatory, antioxidant and protective effects against neurodegenerative mechanisms [55,56]. The depletion of vitamin B3 may be caused by a reduction in several vitamin B3-producing bacteria within the Firmicutes phylum [57], that we found to be linked to PD patients in a previous report [38] and in the present study. On the contrary, the increased levels of cadaverine match well with an overgrown *Escherichia* population; the bacteria in the human gut that can produce this biogenic amine [58]. It has been suggested that dysregulation of cadaverine can be involved in neurodegeneration and the formation of Lewy bodies in PD [59,60]. In vitro studies highlighted that biogenic amines can bind the N-terminal region of the amyloid beta peptide, significantly increasing the aggregation of α-synuclein [61]. Another pathological mechanism by which increased cadaverine levels affect neurodegeneration may be the induction of oxidative stress through the formation of toxic metabolites, including aldehydes, H_2_O_2_ and ammonia [60,62,63]. These pathogenic effects might contribute to promoting an inflammatory environment and α-synuclein aggregation in the ENS [38,60,64]. 

Interestingly, we found an increase in glucuronic acid to be associated with the D phenotype. Multiple mechanisms may underlie the increased levels of glucuronic acid. One of the most reasonable possibilities involves the cleavage of glucuronic acid from glucuronidated xenobiotics by gut β-glucuronidase-bacteria [65]. *Escherichia* are among the opportunistic pathogens in the human gut that harbor a β-glucuronidase enzyme [66,67] and cause the release of glucuronic acid and, consequently, of xenobiotics, such as environmental toxins and drug metabolites with pro-inflammatory effects, that interfere with their inactivation and elimination via the GI tract [65,68]. 

These data are of particular importance since a direct correlation between pesticide exposure and increased risk of PD development has been already reported [69]. Furthermore, two important neurotransmitters generated in the GI tract [70,71] and involved in the regulation of gut motility [72,73], dopamine and serotonin, which are commonly glucuronidated to allow for their easier transport through the body, can be hydrolyzed by β-glucuronidase-bacteria in the GI lumen [68,74]. These observations have led us to hypothesize that increased levels of *Escherichia* and glucuronic acid could be linked to a reduction in glucuronidation, one of the major detoxification pathways [67,75]. This event might contribute to promoting an inflammatory environment in the gut and a reduction in dopamine and serotonin availability, which may be correlated with the more frequent hyperkinetic motor symptoms and GI dysfunctions displayed by D subtypes. 

## 5. Conclusions

Our study highlights that the gut microbiota of PD patients with TD and non-TD motor phenotypes differs in terms of bacterial diversity and taxonomic composition, suggesting a possible relationship between gut dysbiosis and motor impairment. In particular, our findings show that non-TD subtypes were associated with a decrease in bacteria diversity, characterized by a predominant enrichment of Enterobacteriaceae and *Escherichia* and a reduction in Lachospiraceae and other key members. Moreover, interesting modifications of bacterial products, such as a reduction in nicotinic acid and an increase in cadaverine and glucuronic acid, were found to be linked to non-TD phenotype. Although gut microbiota modifications reported in the study could also be ascribed in part to the effect of dopaminergic drugs or to the different stages of disease related to TD and D subtypes, we believe that our findings are of importance and warrant further investigations to clarify the biological interplay between gut microbiota and bacterial metabolite expression in the pathophysiology of PD.

While the PD-causing event and the cause–effect relationship between gut microbiota and the Lewy pathology still remain unclear, the emerging hypothesis, supported also by our findings, suggests that a shift from gut microbial communities to harmful symbionts, probably induced by an exogenous pathogenic insult that gains access to gastric system, might impair the intestinal barrier and initiate the pathological process in the ENS causing inflammation, oxidative stress and α-synuclein aggregation [29,76,77,78]. As the propagation of α-synuclein spreads from gut to brain in a prion-like manner [79], the Lewy pathology advances and the damage to dopaminergic neuron and motor manifestations of the disease increase.

## Figures and Tables

**Figure 1 biomolecules-11-00144-f001:**
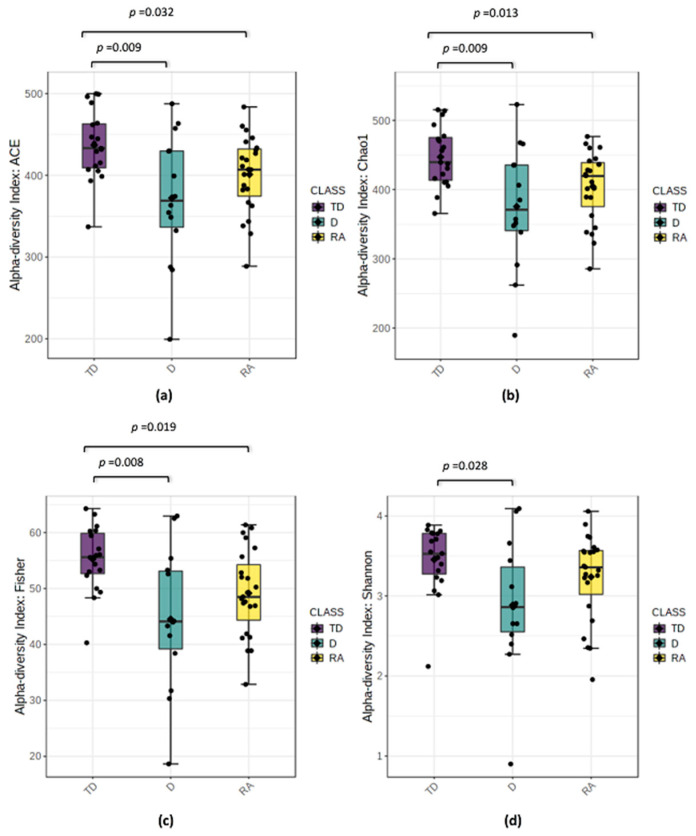
Alpha-diversity box plots throughout different PD phenotypes groups: Tremor Dominant (TD); Akinetic Rigid (AR); Dyskinetic (D). Indices of alpha diversity of the microbial species in the samples: (**a**) Abundance-based Coverage Estimator (ACE), (**b**) Chao, (**c**) Fisher and (**d**) Shannon. *p* values were evaluated using Mann–Whitney U non-parametric test. Median values, interquartile ranges and *p* values ≤ 0.05 were indicated in the plots.

**Figure 2 biomolecules-11-00144-f002:**
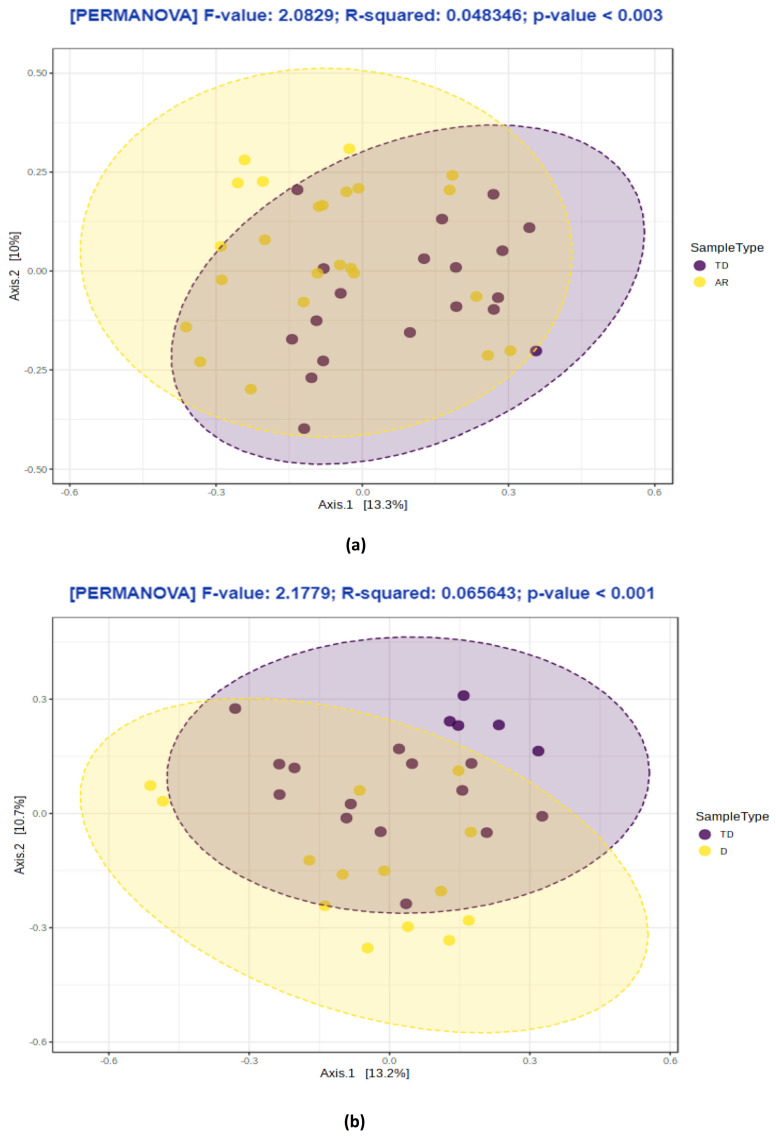
Beta-diversity analysis throughout different PD phenotypes groups: (**a**) Tremor Dominant (TD) vs. Akinetic Rigid (AR); (**b**) Tremor Dominant (TD) vs. Dyskinetic (D). Data are displayed as a 2D plot based on a principal coordinate analysis (PCoA). The statistical significance was evaluated using Permutational Multivariate Analysis of Variance (PERMANOVA).

**Figure 3 biomolecules-11-00144-f003:**
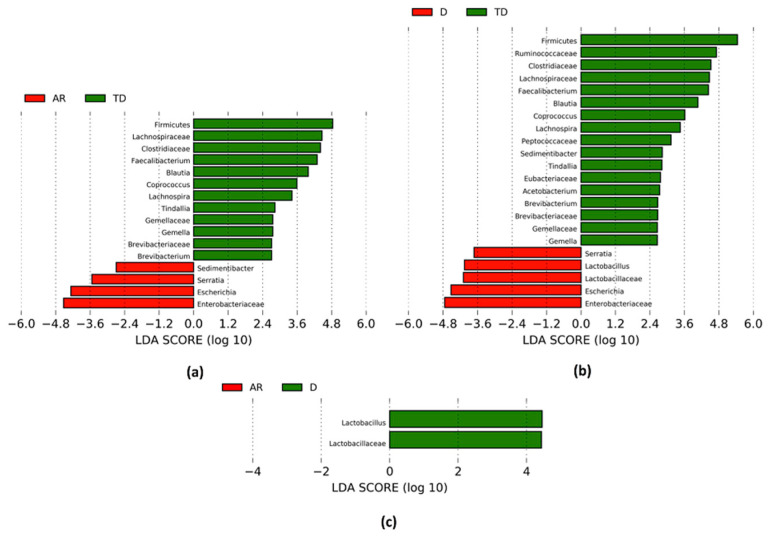
Linear discriminant analysis effect size (LEfSE) analysis: The bar plots represent the significantly different taxa among PD patients with distinct motor phenotypes, based on effect size (Linear discriminant analysis (LDA) score (log 10) > 2). (**a**) Positive LDA score (green) highlights the enriched taxa in PD patients with Tremor Dominant (TD) and negative LDA score (red) shows the enriched taxa in PD patients with Akinetic Rigid (AR); (**b**) Positive LDA score (green) highlights enriched taxa in PD patients with Tremor Dominant (TD) and negative LDA score (red) shows enriched taxa in PD with Dyskinetic (D); (**c**) Positive LDA score (green) highlights enriched taxa in PD patients with Dyskinetic (D) compared to Akinetic Rigid (AR) phenotype; Kruskal–Wallis test (α = 0.05) and Benjamini and Hochberg false discovery rate (FDR) correction test for multiple comparisons were used to evaluate the differences among classes.

**Figure 4 biomolecules-11-00144-f004:**
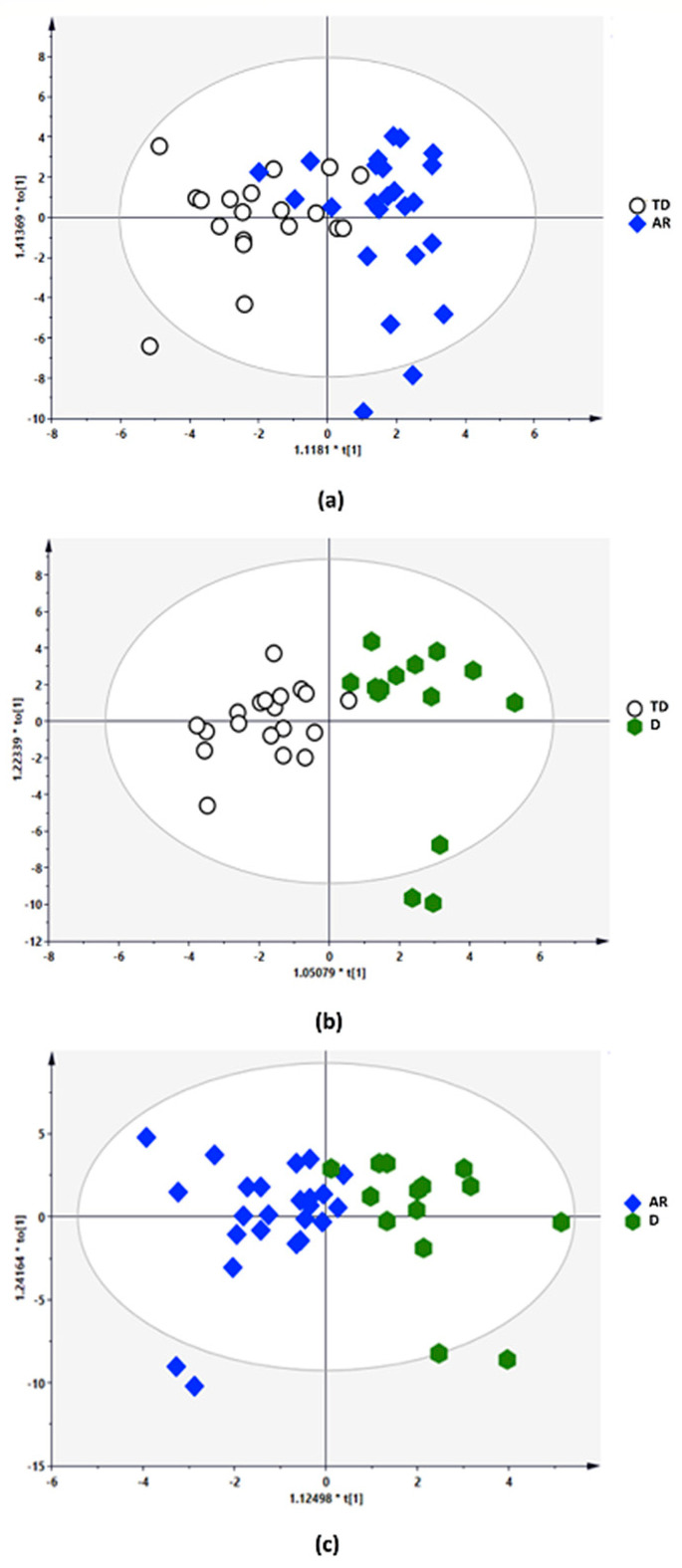
Metabolomic multivariate statistical analysis (MVA): orthogonal partial least-square discriminant analysis (OPLS-DA) score plots of: (**a**) Tremor Dominant (TD) blank circle vs. Akinetic Rigid (AR) blue rhombus; (**b**) Tremor Dominant (TD) blank circle vs. Dyskinetic (D) green hexagon; (**c**) Akinetic Rigid (AR) blue rhombus vs. Dyskinetic (D) green hexagon.

**Figure 5 biomolecules-11-00144-f005:**
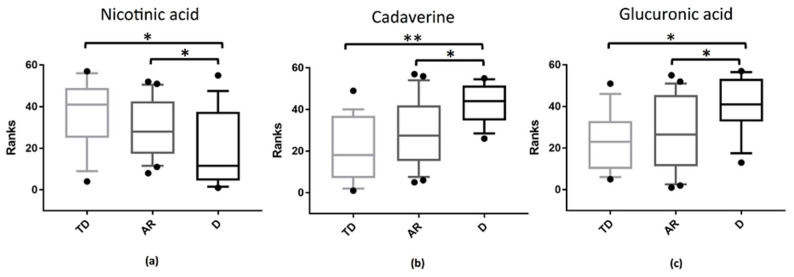
Statistically significant metabolites in the comparison among different PD phenotypes groups: Tremor Dominant (TD); Akinetic Rigid (AR); Dyskinetic (D). Corrected *p* values were evaluated using Mann–Whitney U test followed by Holm–Bonferroni correction test. The metabolites found, (**a**) nicotinic acid, (**b**) cadaverine, (**c**) glucuronic acid, are displayed and expressed in the graphs y axis as ranks (data transformation in which numerical or ordinal values are substituted by their rank once the data are sorted). (*): *p* < 0.05; (**): *p* < 0.01.

**Table 1 biomolecules-11-00144-t001:** Distribution of clinical phenotype groups in PD patients.

Phenotypes	TD	AR	D	*p* Value
				0.434 (TD-AR)
*n* (%)	19 (33.93%)	23 (41.07%)	14 (25.00%)	0.301 (TD-D)
				0.070 (AR-D)

Statistical differences (*p* < 0.05) in the frequency of the Parkinson’s disease (PD) patients in the phenotype groups (TD = Tremor Dominant; AR = Akinetic Rigid; D = Dyskinetic) were determined using chi-square test.

**Table 2 biomolecules-11-00144-t002:** Demographic and clinical characteristics of PD patients based on motor phenotypes.

Variable	PD Patients (TD; *n* = 19)	PD Patients (AR; *n* = 23)	PD Patients (D; *n* = 14)
**Age**, mean ± SD	72.00 ± 8.00	71.00 ± 11.07	68.00 ± 9.21
**BMI**, mean ± SD	27.00 ± 3.93	26.19 ± 3.05	25.61 ± 5.26
**Sex**, *n* (%)			
Male	13 (68.42%)	17 (73.91%)	9 (64.29%)
Female	6 (31.58%)	6 (26.09%)	5 (35.71%)
**Constipation**, *n* (%)			
Yes	9 (52.94%)	9 (40.91%)	9 (64.29%)
No	8 (47.06%)	13 (59.09%)	5 (35.71%)
Missing	2	1	0
**Coffee consumption**, *n* (%)			
Yes	11 (64.71%)	15 (71.43%)	7 (50.00%)
No	6 (35.29%)	6 (28.57%)	7 (50.00%)
Missing	2	2	0
**Smoking status**, *n* (%)			
Yes	1 (5.88%)	0 (0%)	3 (21.43%)
No	16 (94.12%)	19 (100%)	11 (78.57%)
Missing	2	4	0
**Duration of disease**, *n* (%)			
0–3 years	7 (41.18%)	4 (21.05%)	0 (0%)
4–13 years	10 (58.82%)	10 (52.63%)	7 (50.00%)
>13 years	0 (0%)	5 (26.32%)	7 (50.00%)
Missing	2	4	0
**Treatment**, *n* (%)			
Short LD	8 (41.11%)	5 (21.74%)	1 (7.14%)
Long-term LD	9 (47.37%)	11 (47.83%)	6 (42.85%)
L-CIG	2 (10.53%)	7 (30.43%)	7 (50.00%)

PD phenotypes: TD = Tremor Dominant; AR = Akinetic Rigid; D = Dyskinetic; BMI = Body Mass Index.

**Table 3 biomolecules-11-00144-t003:** Significant differences in gut bacteria in PD patients with different motor phenotypes.

Phenotype Comparison	Phylum	Family	Genus	↓/↑	MD	Adjusted *p* Value	
**AR vs. TD**	Actinobacteria	Brevibacteriaceae		↓	−0.417	0.000	* 0.005
			*Brevibacteriun*	↓	−0.417	0.000	* 0.005
	Firmicutes			↓	−0.153	0.005	−
		Clostridiaceae		↓	−0.301	0.008	−
			*Tindallia*	↓	−0.744	0.003	−
		Gemellaceae		↓	−0.286	0.004	−
			*Gemella*	↓	−0.286	0.004	−
		Lachnospiraceae		↓	−0.366	0.004	* 0.005
			*Blautia*	↓	−0.344	0.003	* 0.014
			*Coprococcus*	↓	−0.931	0.002	* 0.014
			*Lachnospira*	↓	−0.450	0.005	* 0.036
		Ruminococcaceae	*Faecalibacterium*	↓	−0.403	0.009	−
		Peptostreptococcaceae	*Sedimentibacter*	↑	0.424	0.000	* 0.004
	Proteobacteria	Enterobacteriaceae		↑	0.819	0.005	* 0.010
			*Escherichia*	↑	1.039	0.002	−
			*Serratia*	↑	0.925	0.002	* 0.026
**D vs. TD**	Actinobacteria	Brevibacteriaceae		↓	−0.481	0.000	* 0.004
			*Brevibacteriun*	↓	−0.481	0.000	* 0.004
	Firmicutes			↓	−0.215	0.002	−
		Clostridiaceae		↓	−0.454	0.002	−
			*Tindallia*	↓	−0.811	0.012	−
		Eubacteriaceae		↓	−0.199	0.002	* 0.003
			*Acetobacterium*	↓	−0.422	0.002	* 0.008
		Gemellaceae		↓	−0.394	0.000	* 0.040
			*Gemella*	↓	−0.394	0.000	* 0.040
		Lachnospiraceae		↓	−0.422	0.003	* 0.008
			*Blautia*	↓	−0.407	0.004	* 0.016
			*Coprococcus*	↓	−0.888	0.003	* 0.036
			*Lachnospira*	↓	−0.740	0.000	* 0.005
		Lactobacillaceae		↑	0.095	0.002	−
			*Lactobacillus*	↑	0.841	0.002	−
		Peptococcaceae		↓	−0.450	0.000	* 0.037
		Peptostreptococcaceae	*Sedimentibacter*	↓	−0.893	0.000	* 0.000
		Ruminococcaceae		↓	−0.487	0.002	−
			*Faecalibacterium*	↓	−0.493	0.004	−
	Proteobacteria	Enterobacteriaceae		↑	1.061	0.003	−
			*Escherichia*	↑	1.273	0.003	* 0.019
			*Serratia*	↑	1.062	0.004	* 0.010
**D vs. AR**	Firmicutes	Lactobacillaceae		↑	0.814	0.008	−
			*Lactobacillus*	↑	0.749	0.007	−

Kruskall–Wallis test and Analysis of Covariance (ANCOVA) performed using Generalized Linear Model (GLM) followed by the Benjamini and Hochberg false discovery rate (FDR) and Bonferroni correction test for multiple comparisons in SPSS (version 25.0 for Windows); MD: Mean difference between logarithmic value of relative abundance in the different phenotype groups TD (*n* = 19): Tremor Dominant, AR (*n* = 23): Akinetic Rigid, and D (*n* = 14): Dyskinetic; adjusted *p* values (*p* < 0.05) obtained by Kruskall–Wallis and FDR correction tests; * adjusted *p* values (*p* < 0.05) obtained by ANCOVA and Bonferroni correction test where sex, age, BMI, constipation, coffee consumption, smoking status and pharmacological treatment were covariates; (−): adjusted *p* values (*p* > 0.05); ↓ Significantly reduced, ↑ Significantly increased with respect to first phenotype shown.

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
