# Peer review of "Clinical Phenotypes of Parkinson’s Disease Associate with Distinct Gut Microbiota and Metabolome Enterotypes"

_biomolecules, 2021, doi:10.3390/biom11020144_

Round 1
Reviewer 1 Report
This is a subject that has acquired high significance in understanding of the pathogenesis of PD. This manuscript details a standard approach to understanding changes in microbiome in different PD clinical conditions. The findings are similar to some others though differences are expected due to different populations being studied.
However, the manuscript is very confusing in the message and conclusions that are being presented. Most of the results are focused on a multitude of statistical analyses but interpretation of the findings is thin as it relates to our understanding of how the bacterial flora can have an effect on disease progression. Maybe there are no conclusions to be made but could the authors please try and make a tighter link between the biota changes and the clinical phenotypes. There is over reliance on statistics without discussion of biological significance. Because there is a significant p value, it must mean something to the disease while it might be coincidental. We are still left with the problem that does the disease affect the flora or does the flora affect the disease. There are no human data to support this. If it is the latter, what is the mechanism. One could consider that the bacteria are causing peripheral inflammation leading to SN toxicity, or as suggested that the bacteria are affecting the aggregation of alpha synuclein with the downstream consequences.
1. The figures are hard to view even at 2x on pdf. Several of the figures are upside down.
2. Please discuss the features of the bacterial families being assessed. It is insufficient to conclude that Enterobacteria are proinflammatory when everyone has E. Coli in their gut but maybe different strains. All gram negative bacteria will have lipopolysaccharides of some form.
3. The metabolic measurements are limited by small numbers of samples. Could the authors explain why it is believed, for example, that increases in gut cadaverine could have consequences on PD neurodegeneration.
4. The concept that maybe PD starts in the gut is wide spread but there are dissenting opinions. Maybe a balancing reference should be included from a neuropathologist who think it could start in olfactory bulb.
5. The over arching hypothesis of many in this field is that the metabolism of L-dopa can be affected by the gut flora, and this will affect the effectiveness and therefore clinical manifestations of the disease. This feature should be discussed and referenced; l do not see listed the key publications in this area.
Author Response
Rebuttal to comments of Reviewers
We have reworked the manuscript according to the Reviewers’ comments and suggestions.
In the revised manuscript the changes made are highlightedwith “Track Changes” of Word software.
Comments and Suggestions for Authors
Reviewer 1
This is a subject that has acquired high significance in understanding of the pathogenesis of PD. This manuscript details a standard approach to understanding changes in microbiome in different PD clinical conditions. The findings are similar to some others though differences are expected due to different populations being studied.
However, the manuscript is very confusing in the message and conclusions that are being presented. Most of the results are focused on a multitude of statistical analyses but interpretation of the findings is thin as it relates to our understanding of how the bacterial flora can have an effect on disease progression. Maybe there are no conclusions to be made but could the authors please try and make a tighter link between the biota changes and the clinical phenotypes. There is over reliance on statistics without discussion of biological significance. Because there is a significant p value, it must mean something to the disease while it might be coincidental. We are still left with the problem that does the disease affect the flora or does the flora affect the disease. There are no human data to support this. If it is the latter, what is the mechanism. One could consider that the bacteria are causing peripheral inflammation leading to SN toxicity, or as suggested that the bacteria are affecting the aggregation of alpha synuclein with the downstream consequences.
R.We thank the reviewer for her/his commentsand suggestions. We providethe following possible explanation for the potential mechanisms in the Conclusion section: “While the PD-causing event and the cause-effect relationship between gut microbiota and disease still remain unclear, the emerging hypothesis, supported also by our findings, suggests that a shift of gut microbial communities versus harmful symbionts, probably induced by an exogenous pathogenic insult that gains access to gastric system, might impair the intestinal barrier and initiate the pathological process in the ENS causing inflammation, oxidative stress and a-synuclein aggregation. As the propagation of a-synuclein spreads from gut to brain in a prion-like manner, the Lewy pathology advances and the damage to dopaminergic neurons and the motor manifestations of the disease increase” (pag 12, lines 787-803).
- The figures are hard to view even at 2x on pdf. Several of the figures are upside down.
R:We had some problems uploading the figures in the text. For this reason, following the Reviewer's suggestions we uploaded the figures as high resolution TIFF format in a separate zipped file, as requested in the author’s guidelines.
- Please discuss the features of the bacterial families being assessed. It is insufficient to conclude that Enterobacteria are proinflammatory when everyone has E. Coli in their gut but maybe different strains. All gram negative bacteria will have lipopolysaccharides of some form.
R: We recognize that the discussion about this point was not accurate.
We provided the following explanation: “Indeed, the gut microbiota dysbiosis is characterized by a shift in relative bacterial abundances with a prevalence of pathobionts belonging to the phylum Proteobacteria (i.e. Enterobacteriaceae), while beneficial symbionts belonging mainly to phylum Firmicutes and Bacteroides are less expressed. This persistent imbalance of gut microbial community between harming and non-harming symbionts induces an immune reaction, which promotes an inflammation status that represents a favourable microenvironment for the growth of the same Enterobacteriaceae ” (pag 10, lines 679-684).
- The metabolic measurements are limited by small numbers of samples. Could the authors explain why it is believed, for example, that increases in gut cadaverine could have consequences on PD neurodegeneration.
R: We comply with this Reviewer point and we provided the following explanation:
“It has been suggested that a dysregulation of cadaverine can be involved in neurodegeneration and formation of Lewy bodies in PD. Invitro studies highlighted that biogenic amines could bind the N-terminal region of the amyloid beta peptide, significantly increasing the aggregation of α-synuclein [63]. Another pathological mechanism by which increased cadaverine levels affect the neurodegeneration may be induction of oxidative stress throughthe formation of toxic metabolites, including aldehydes, H2O2 and ammonia. These pathogenic effects might contribute promoting an inflammatory environment and α-synuclein aggregation in the ENS” (pag 11, lines 749-756).
- The concept that maybe PD starts in the gut is wide spread but there are dissenting opinions. Maybe a balancing reference should be included from a neuropathologist who think it could start in olfactory bulb.
R: We followed the Reviewer's suggestions and we discussed and cited the references in the text: “On the other hand, different lines of evidence have suggested that α-synucleinopathy could originate in the olfactory bulb or in the brain, and then spread to the peripheral autonomic nervous system” (pag 2, lines 70-72).
- The over arching hypothesis of many in this field is that the metabolism of L-dopa can be affected by the gut flora, and this will affect the effectiveness and therefore clinical manifestations of the disease. This feature should be discussed and referenced; l do not see listed the key publications in this area.
R: We followed the Reviewer's suggestions and we discussed and cited the references in the text: “however, several points of interest remain to be established. One of these concerns whether the dopaminergic drugs can affect gut bacteria or vice versa. Recent studies highlighted that gut microbes can affect the metabolism of L-DOPA, the most effective anti-Parkinson medication, potentially reducing its effectiveness and therefore the clinical manifestations of the disease”(pag 2, lines 74-77)
Reviewer 2 Report
Sarah et al. tried to determine the association between distinct gut microbiota and metabolome enterotypes with heterogeneity of Parkinson's disease (PD). It is a great idea, however, the results presented in the current manuscript cannot support the hypothesis. The data resolution is too low to make any conclusion. Some words are reversed and some figures are missing. The number for the statistical analysis is too limited and the power analysis is also missing. There are so many details are missing and I am wondering the results are the same, not convincing to their attitudes to the manuscript. I would suggest the senior authors carefully read the manuscript before submission and find the biostatistician to get the reasonable results to support their hypothesis. I will suggest to reject this manuscript.
Author Response
Response to Reviewer 2 Comments
Sarah et al. tried to determine the association between distinct gut microbiota and metabolome enterotypes with heterogeneity of Parkinson's disease (PD). It is a great idea, however, the results presented in the current manuscript cannot support the hypothesis. The data resolution is too low to make any conclusion. Some words are reversed and some figures are missing. The number for the statistical analysis is too limited and the power analysis is also missing. There are so many details are missing and I am wondering the results are the same, not convincing to their attitudes to the manuscript. I would suggest the senior authors carefully read the manuscript before submission and find the biostatistician to get the reasonable results to support their hypothesis. I will suggest to reject this manuscript.
Response:We are quite amazed by the unflattering judgment expressed by this reviewer. We have worked hard to make the study hypothesis understandable and to try to draw consistent conclusions, although they cannot represent any certainty in a field not yet fully explored. We honestly thought our study could further contribute to the ongoing scientific debate on this topic. We are convinced that the opinions of the other reviewers have helped, we hope, improving the manuscript and if this reviewer would like to make a further assessment based on the changes made, we would be extremely grateful to her/him in advance.
Reviewer 3 Report
The study by Vascellari et al explores the differences in the gut microbiome and metabolome between Parkinson’s disease patients with varying phenotypes, namely tremor dominant, or non-tremor dominant with either akinetic rigidity or dyskinesia. The study provides an important look into relationship between the microbiome and the pathophysiological motor differences observed with Parkinson’s disease. There are, however, several items that should be addressed.
- There are several issues with the figures. All of the figures are very low resolution. This may be a website/upload issue, but please make sure the figures are of sufficient size and resolution prior to uploading. Some figures were unreadable. Figures 1 and 2 are upside down and Figure 3 is too big for the page, so parts are missing.
- In line 78 of the methods, the authors should detail what institute the approving “local” ethics committee was part of.
- Lines 84-86: Please clarify this sentence. It is unclear what “at least 2/4 in scores 4.32-4.33-4.34 in UPDRS IV” and “<3 aa” mean.
- Please define alpha- and beta-diversity.
- Consider combining tables 3 and 4
- Figure 3 is difficult to understand. This is in part because of the issues already mentioned with the figure, but also because of the figure legend. The figure shows red as AR and green as TD, but the figure legend describes green as positive and red as negative. It is also unclear whether panel c is cutoff or not, but there are no bars shown for AR as detailed in the legend. The legend also details acronyms that are not shown in the figure (ex: AR and TD). As this is an important figure, please fix the figure and clarify the legend.
- In figure 4 please add the color designations to the legend.
- In the discussion, the authors repeatedly use the term “severe” (ex: the less severe TD phenotype). They did not discuss disease severity as a variable in the paper, so it is confusing on whether they mean the severity within a phenotype based on UPDRS scores or are simply comparing the TD to non-TD or non-TD AR to D phenotypes. If it is the latter, this is a qualitative difference in types of symptoms and should not be described as more or less severe.
- The paper should be carefully proof read as there are many typographical and grammatical errors throughout.
Author Response
Response to Reviewer 3 Comments
The study by Vascellari et al explores the differences in the gut microbiome and metabolome between Parkinson’s disease patients with varying phenotypes, namely tremor dominant, or non-tremor dominant with either akinetic rigidity or dyskinesia. The study provides an important look into relationship between the microbiome and the pathophysiological motor differences observed with Parkinson’s disease. There are, however, several items that should be addressed.
Point 1.There are several issues with the figures. All of the figures are very low resolution. This may be a website/upload issue, but please make sure the figures are of sufficient size and resolution prior to uploading. Some figures were unreadable. Figures 1 and 2 are upside down and Figure 3 is too big for the page, so parts are missing.
Response 1:We had some problems uploading the figures in the text. For this reason, following the Reviewer's suggestions we uploaded the figures as high resolution TIFF format in a separate zipped file, as requested in the author’s guidelines.
Point 2.In line 78 of the methods, the authors should detail what institute the approving “local” ethics committee was part of
Response 2:We followed the Reviewer's suggestions and we added the information concerning the local Ethical Committee Institute: “All patients provided written informed consent after the study was approved by the local Institutional Ethics Committee (Prot.PG/2017/17817) of the Azienda Opedaliero Universitaria di Cagliari, Italy” (pag 3, lines 102-103).
Point 3: Lines 84-86: Please clarify this sentence. It is unclear what “at least 2/4 in scores 4.32-4.33-4.34 in UPDRS IV” and “<3 aa” mean
Response 3:We thank the reviewer for this comment. MDS-Unified Parkinson's Disease Rating Scale (MDS-UPDRS) is a universally approved rating scale for PD. The scale evaluates various aspects of Parkinson’s disease including non-motor and motor experiences and it can be used in a clinical setting as well as in research. It is divided in 4 parts, with the 4th session dedicated to the evaluation of dyskinesia. In our paper we defined patients as "dyskinetic" if they reached a score of >2 over 4 in the sequences 4.2, 4.3, 4.4 of the UPDRS scale. We modified the text to a clearer version and we add the link to the UPDRS Scale update version for better understanding (pag 3, lines 106-113).
Point 4: Please define alpha- and beta-diversity.
Response 4:We defined alpha- and beta-diversity in the text: “The variation of microbial communities within sample (alpha-diversity) and between samples (beta-diversity) was estimated in the PD phenotype groups”(pag 6, lines 437- 438).
Point 5: Consider combining tables 3 and 4
Response 5:We followed the Reviewer's suggestions and we combined Tables 3 and 4. In the new Table 3 we inserted the “ * “ symbol and the respectivesp values to indicate the significantly abundance of bacteria after the correction for several confounding factors obtained by Analysis of Covariance (ANCOVA), followed by Bonferroni correction for multiple comparison. We apologize because in the old Table 4 there was a mistake. The column indicating the mean differences (MD) between logarithmic values of relative abundance in the different phenotype groups was wrongly indicated under “p value”.
Point 6: Figure 3 is difficult to understand. This is in part because of the issues already mentioned with the figure, but also because of the figure legend. The figure shows red as AR and green as TD, but the figure legend describes green as positive and red as negative. It is also unclear whether panel c is cutoff or not, but there are no bars shown for AR as detailed in the legend. The legend also details acronyms that are not shown in the figure (ex: AR and TD). As this is an important figure, please fix the figure and clarify the legend.
Response 6: We agree with Reviewer's suggestions and we fixed the figure and clarified the legend. Concerning the panel c the figure of the linear discriminant analysis effect size (LEfSe), obtained using Galaxy tool, shows only the bars of the bacteria significantly increased in the Dyskinetic phenotype (i.e. Lactobacillaceae and Lactobacillus) compare to Akinetic Rigid. We corrected the legend and replaced the name of phenotype groups with the acronyms in the figure (pag 8, lines 519- 526).
Point 7:In figure 4 please add the color designations to the legend.
Response 7: We added the color designations to the legend in the figure 4 (pag 9, lines 614- 616), following the Reviewer's suggestions.
Point 8: In the discussion, the authors repeatedly use the term “severe” (ex: the less severe TD phenotype). They did not discuss disease severity as a variable in the paper, so it is confusing on whether they mean the severity within a phenotype based on UPDRS scores or are simply comparing the TD to non-TD or non-TD AR to D phenotypes. If it is the latter, this is a qualitative difference in types of symptoms and should not be described as more or less severe
Response 8:We thank the reviewer for this comment. The ambiguity of the terminology can be explained by the fact that non-TD phenotypes are proposed to be associated to a worse prognosis and a faster progression of PD compared to TD patients. We agree with the reviewer that our writing was confusing, consequently we deleted the referral to "more severe" phenotypes and we added a few more sentences to clarify this concept (pag 10, lines 709- 713).
Point 9:The paper should be carefully proof read as there are many typographical and grammatical errors throughout.
Response 9: We revised typographical and grammatical form of the manuscript.
Round 2
Reviewer 2 Report
The authors tried to determine the association between clinical phenotype and distinct gut microbiota and metabolome enterotypes. The novelty is still high. In the revised manuscript, the word and figure format have been significantly improved, which is appreciated.
The number for the statistical analysis is still a concern, some figures only have sample number is 2. How these limited samples can provide the significance? In another, does the association is specific with TD/non-TD, or the gut results are also associate with others, such as cognition. PD is progressive, thus different stages may have dynamic gut results. Have the authors considered this factor in the correlation analysis?
To determine the association, have the authors normalized with genetic and pathological background, since GBA, LRRK2, SNCA, and pathological (abeta, tau, TDP43, and their PTM forms) proteinopathy may reversely interfere the gut results. If the authors didn't have the results, the dicusssion for the weakness of the manuscript is needed.
I would suggest a major revision for this manuscript.
Author Response
Rebuttal to comments of Reviewers
We have reworked the manuscript according to the Reviewer 2’ comments and suggestions.
In the revised manuscript the changes made are highlighted with “Track Changes” of Word software.
Response to Reviewer 2 Comments
The authors tried to determine the association between clinical phenotype and distinct gut microbiota and metabolome enterotypes. The novelty is still high. In the revised manuscript, the word and figure format have been significantly improved, which is appreciated.
Point 1: The number for the statistical analysis is still a concern, some figures only have sample
number is 2. How these limited samples can provide the significance?
Response 1: We are extremely grateful to the Reviewer 2 for making a further assessment based on the changes made.
We comply with this Reviewer point and we provided to remove the p values in Table 2 and the related statistical significance in the Result section (pag 6, line 419).
Point 2: In another, does the association is specific with TD/non-TD, or the gut results are also associate with others, such as cognition. PD is progressive, thus different stages may have dynamic gut results. Have the authors considered this factor in the correlation analysis?
Response 2: In the study we didn’t consider the correlation between gut microbiota changes and cognitive impairment in PD patients with different motor phenotypes.However, because we recognize that the different stages of disease may also reflect dynamic gut results, we had already considered this point in the Conclusion section: “Although gut microbiota modifications reported in the study could also be ascribed in part to the effect of dopaminergic drugs or to the different stages of disease related to TD and D subtypes, we believe that our findings are of importance and warrant further investigations to clarify the biological interplay between gut microbiota and bacteria metabolites expression in the pathophysiology of PD” (pag 16, lines 768-782).
Point 3: To determine the association, have the authors normalized with genetic and pathological background, since GBA, LRRK2, SNCA, and pathological (abeta, tau, TDP43, and their PTM forms) proteinopathy may reversely interfere the gut results. If the authors didn't have the results, the dicusssion for the weakness of the manuscript is needed.
Response 3: We comply with this Reviewer point and we provided the following integrations in the Discussion and Conclusion sections:
“Even though more genetic and neuropathological studies are needed to establish the relationship of the gut microbiota changes with α-synuclein, leucine-rich repeat kinase 2, or glucocerebrosidase gene expression, as well as with other proteinopathies (i.e amyloid-beta, tau and TAR DNA binding protein 43), our results suggest that gut dysbiosis may play a role in the different natural history and prognosis of disease related to TD and non-TD phenotypes” (pag 15, lines 684-688).
“While the PD-causing event and the cause-effect relationship between gut microbiota and Lewy pathology still remain unclear, ..” (pag 17, lines 783-784).
